# Linker Functionalization Strategy for Water Adsorption in Metal–Organic Frameworks

**DOI:** 10.3390/molecules27092614

**Published:** 2022-04-19

**Authors:** Rafaela Maria Giappa, Anastasios G. Papadopoulos, Emmanuel Klontzas, Emmanuel Tylianakis, George E. Froudakis

**Affiliations:** 1Department of Chemistry, University of Crete, Voutes Campus, GR-71003 Heraklion, Greece; chemp956@edu.chemistry.uoc.gr; 2Theoretical and Physical Chemistry Institute, National Hellenic Research Foundation, GR-11635 Athens, Greece; anastp@eie.gr (A.G.P.); klontzas@eie.gr (E.K.); 3Department of Materials Science and Technology, University of Crete, Voutes Campus, GR-71003 Heraklion, Greece; tilman@materials.uoc.gr

**Keywords:** metal–organic frameworks, water, water harvesting, functionalization, ab initio, Monte Carlo

## Abstract

Water adsorption in metal–organic frameworks has gained a lot of scientific attention recently due to the potential to be used in adsorption-based water capture. Functionalization of their organic linkers can tune water adsorption properties by increasing the hydrophilicity, thus altering the shape of the water adsorption isotherms and the overall water uptake. In this work, a large set of functional groups is screened for their interaction with water using ab initio calculations. The functional groups with the highest water affinities form two hydrogen bonds with the water molecule, acting as H-bond donor and H-bond acceptor simultaneously. Notably, the highest binding energy was calculated to be −12.7 Kcal/mol for the -OSO_3_H group at the RI-MP2/def2-TZVPP-level of theory, which is three times larger than the reference value. Subsequently, the effect of the functionalization strategy on the water uptake is examined on a selected set of functionalized MOF-74-III by performing Monte Carlo simulations. It was found that the specific groups can increase the hydrophilicity of the MOF and enhance the water uptake with respect to the parent MOF-74-III for relative humidity (RH) values up to 30%. The saturation water uptake exceeded 800 cm^3^/cm^3^ for all candidates, classifying them among the top performing materials for water harvesting.

## 1. Introduction

With two thirds of the Earth’s surface covered by water and thirteen thousand trillion liters in the form of vapor and droplets in the atmosphere [1], it is evidently clear that water is ubiquitous and of paramount importance for many physical and chemical functions in nature. As the population of the earth is growing, the demand for larger amounts of fresh water is expected to grow rapidly, especially in arid areas, which are expected to expand [2]. Sorption-based harvesting of atmospheric water has attracted a lot of attention due to the low requirements for energy as well as its operation in low relative humidity areas to produce potable water [3]. Water-based sorption systems can also be utilized for reducing the rapidly increasing energy demands and lowering the corresponding carbon footprint at relatively low operational cost [4]. Apart from being the most abundant molecule, water has drawn a lot of scientific and technological attention due to its chemical and physical complexity, which mainly arise from the formation of hydrogen bonds; their strong directionality is responsible for a remarkable set of anomalous physical and chemical properties, both in the bulk state and under confinement. Exhibiting an intriguing array of unusual properties, the elucidation of its behavior and the mastering of its processes in various experimental conditions has been a great scientific challenge with many remaining open pathways of pursuit.

A major area of study closely associated with the peculiar properties of water is its adsorption by porous solids because the latter are involved in a variety of industrial processes. Apart from the conventional porous solids such as zeolites [5,6], mesoporous silicas [7] and carbon-based materials [5,8], metal–organic frameworks (MOFs) have attracted substantial attention in the last two decades due to their unique properties. MOFs are a relatively new class of porous crystalline material consisting of inorganic metal ions or metal clusters linked by organic ligands through coordination bonds, and which stand out for their ultrahigh porosity (up to 90% free volume) and extended internal surface areas (beyond 6000 m²/gr) [9]. Stretching the limits of their physical properties, MOFs’ composition, structure, functionality and pore metrics can be varied by choosing appropriate building block components following the reticular chemistry approach [10]. As a result, a large variety of assembled MOFs has been reported in the literature, which holds a lot of promise for exceptional performance in various research fields [11]. However, due to the lability of ligand–metal bonds, their most significant drawback is their affinity towards water that renders mechanical and chemical instability. Most of the earlier reported MOFs are sensitive to water, exhibiting ligand displacements and phase changes and/or structural decomposition when the environment contains moisture [6,12,13]. This fact constitutes a major problem that needs to be tackled since water molecules in moisture form are present in many MOF applications, not to mention in their preparation and synthesis conditions.

Some areas of application of water-stable MOFs include selective adsorption [14], membrane separation [15], sensing [16], catalysis [17] and proton conduction [18], whereas applications directly requiring the capture and release of water are adsorption-based atmospheric water harvesting (AWH) [19], adsorption-driven heat pumps (ADHPs) and chillers (ADCs) [20], sorptive thermal batteries [21] and adsorptive cooling desalination [22]. Hence, thorough understanding of the aspects that make a MOF structure robust under ambient humidity and moisture is of utmost importance. In addition, the adsorbent performance is equally important and should meet the different operational requirements [19,23] that have been established for each of the abovementioned applications involving adsorption and desorption of water. Therefore, as soon as the water adsorption isotherm for a candidate MOF is obtained, it can be proposed for a targeted application. Relevant engineering aspects should also be considered during the design of a water-based adsorption process [24]. Regarding the factors that influence the performance of MOFs for water adsorption, pore structure (i.e., size distribution, shape, dimensionality, etc.) and chemical composition (i.e., open metal sites, size of the linkers, functionalities, defects, etc.) can effectively tune their hydrophilicity, affecting the uptake and the type of the isotherm that is usually observed [13]. Open metal sites can also increase the hydrophilicity of these materials since they act as strong binding sites for water; however, depending on the type of the metal, this very effect might prevent the release of bound water during desorption. The introduction of hydrophilic or hydrophobic functional groups to the organic linkers can further tune the hydrophilicity of MOFs and in some cases, the introduction of hydrophobic groups have been shown to decrease the degradation of water-unstable MOFs. As demonstrated in the case of CAU-10-X [25,26], water adsorption characteristics of CAU-10 are enhanced by the decoration of its porous walls by different functional groups. Depending on the hydrophobic/hydrophilic nature of the substituents, H-bonding interactions and accessibility of the hydrophilic μ-OH functionalities define the shape of the adsorption isotherm and the point of water condensation, namely its water uptake behavior. Molecular simulations performed to investigate the water sorption properties of a series of functionalized UiO-66-X (X: -SO_3_H, -2COOH, -NH_2_ and -Br) [27] showed that the introduction of the hydrophilic groups increased the enthalpy of the adsorption of water in the MOF structures and caused a shift in the condensation pressure to lower relative pressure values. Water sorption studies of functionalized derivatives of the mesoporous MIL-101 [28,29,30] indicated that this MOF candidate can adsorb large amounts of water, with controllable trapping pressure of the water adsorption depending on the functional group, since even partial functionalization of the parent structure induces large changes in the corresponding water adsorption properties.

Induced by the fact that many MOF structures have a phenyl group as an organic linker, and taking into account from previous studies that introducing various different organic functional groups onto the organic ligands of MOFs [31,32,33,34] can tune guest molecule uptake, we conducted a series of ab initio and density functional theory (DFT) calculations on the interactions of water with strategically selected functionalized benzene molecules. The study of the interaction of aromatic molecules with water is of paramount importance and in need of thorough investigation as binding in these complexes has a complicated yet not fully understood nature. Aiming to shed light on the nature of these interactions, we herein report the calculated binding energies between a water molecule and a series of functionalized benzenes (FBs), pointing out the dimers with the strongest interactions in an attempt to obtain more targeted functionalization of the organic moieties in MOF structures. In order to check the effectiveness of the functional groups to enhance water affinity in MOFs, Grand Canonical Monte Carlo simulations were employed at room temperature and at a range of relative humidity levels in order to calculate the water isotherms for a selected set of functional groups grafted to the linkers of Mg-MOF-74-III, a member of a series of isoreticularly expanded structures with larger pore apertures based on the widely studied Mg-MOF-74 structure [35]. Mg-MOF-74-III is the third member (-III) of a series of isoreticularly expanded structures with larger pore apertures based on the widely studied Mg-MOF-74 structure. This structure preserves the same topology and the same 1D channels with a hexagonal shape aligned along the infinite metal oxide chains, but the pore diameter becomes increased (22 Å in -III compared to 10 Å in -I) due to the expansion of the linker length by including three (-III) phenylene rings instead of the single (-I) ring of the parent structure. The criteria for selecting this MOF are summarized by its stability after exposure to air [35], its ability to modify the organic linkers with functional groups [36,37,38], the existence of open metal sites that can effectively bind water, its pore size due to the extended linker that allows the accommodation of the bulky functional groups considered in this study (i.e., without overlapping with other parts of the structure) and the sufficient pore volume to accommodate a considerable amount of water molecules. Finally, this structural analog—along with its functionalized versions—contains pores with sizes below the critical diameter D_c_ of the working fluid (i.e., 20.76 Å for water at 25 °C), a characteristic that might possibly serve to avoid the undesirable hysteresis that is present during the desorption of water [39].

## 2. Methodology

### 2.1. Quantum Chemical Calculations

A diverse set of 40 functional groups was tested for the strength of interaction with H_2_O. Benzene was considered the parent linker, since many of the organic linkers used in MOFs are based on aromatic rings and functionalized benzenes (FBs) comprise the simplest viable model to study the interaction of water with this set of functional groups. The functional groups were selected based on our chemical intuition and findings from similar studies in the past [32,33,34] and were introduced to the organic linker model by substituting a hydrogen atom from the benzene ring. The geometries of the FBs were optimized using second-order Moller−Plesset (MP2) perturbation theory in the resolution of identity (RI) approximation [40], along with the def2-TZVPP [41] basis set and the corresponding MP2 optimized auxiliary basis set for the RI approximation. The convergence criterion for the Hartree–Fock self-consistent field (SCF) was set to be 10^−8^ au. No symmetry constraints were set and numerical frequency calculations were performed so as to verify the optimized structures as global minima. In order to identify the most stable conformer in terms of electronic energy, several initial conformations of the FBs were created by adjusting the specific bond angles and dihedrals of the functional groups. Subsequently, the interaction of an H_2_O molecule with the corresponding functional groups was tested for the optimized geometries of the FBs. To ensure that the global minimum of each H_2_O–FB dimer is identified, a large number of intermolecular conformers were initially created by screening many different potential binding sites and orientations of the H_2_O molecule around the FB. All H_2_O–FB conformers were allowed to fully relax at the same level of theory described above. The binding energy was calculated according to the formula B.E. = E_dimer_ − E_linker_ − E_H2O_, where B.E. is the binding energy, E_dimer_, E_linker_ and E_H2O_ are the energies of the optimized dimer, functionalized linker and H_2_O molecule, respectively. The energy of the dimer, E_dimer_, was corrected by applying the counterpoise correction for the basis-set superposition error (BSSE), according to the formula proposed by Boys and Bernardi [42]. All calculations were performed using Turbomole software package [43] and the optimized dimer geometries were visualized with Chemcraft [44] graphical program.

### 2.2. Monte Carlo Simulations

Monte Carlo simulations in the Grand Canonical Ensemble (GCMC) were employed to calculate the water uptake as a function of humidity for a selected subset of functional groups introduced on the organic linkers of Mg-MOF-74-III [35]. The crystal structure of Mg-MOF-74-III that was used during the simulations has been reported in the original paper describing the synthesis and characterization of the whole series of expanded Mg-IRMOF-74 analogs [35]. The corresponding lattice parameters of the conventional cell were a = b = 45.8395 Å, c = 6.4739 Å, α = β = 90.0° and γ = 120.0°. The lattice parameters and the positions of atoms belonging to the original or the functionalized MOF structures were kept fixed during simulations. The functionalized structures were built by substituting each methyl group located on the middle phenyl ring with the functional groups. Following the trend obtained by the ab initio calculations, we selected 3 functional groups that bind H_2_O molecules strongly, namely -OSO_3_H, -CONH_2_ and -C(OH)_3_.

H_2_O molecules were represented using the SPC/E model [45] according to which three interaction points corresponding to the three atoms of the water molecule are used. Bond lengths between hydrogen and oxygen atoms of the water were kept fixed at 1.0 Å and the H–O–H bond angle at 109.47°. The MOF structure was kept rigid during simulations while water molecules were allowed to translate and rotate. Interactions between guest molecules and between guest molecules and the atoms of the host material were taken into account. Van der Waals interactions were described by using the Lennard–Jones potential. In order to accurately describe the non-bonding interactions, the parameters ε and σ of the potential were calculated for the framework atoms by fitting the quantum mechanics (QM)-derived potentials to the Lennard–Jones function; the fitting procedure is presented in the Results section. For the framework atoms, the values of the parameters ε and σ were taken from the Dreiding force field [46]. Lorentz–Berthelot mixing rules were used for the calculation of cross parameters between different pairs of atoms. Regarding the electrostatic interactions, Coulomb interactions were taken into account by calculating the partial atomic charges and were handled by the Ewald summation technique [47]. The partial charges of water molecules were taken from the SPC model, while for framework atoms they were calculated with the CHELPG method [48] at the RI-MP2/def2-TZVPP-level of theory, with the Gaussian software package [49]. Charges were calculated using the cluster approximation, i.e., two molecular fragments were created from the periodic crystal structure of Mg-MOF-74-III, one containing the metal corner with the magnesium atoms and the other the desired linker. The total charge of the framework was kept neutral by applying small adjustments to the partial charges of the framework atoms. In all cases, periodic boundary conditions were applied to extrapolate the results to the bulk values. For the Van der Waals interactions, a cut-off distance of 12.8 Å was used, considering no tail corrections beyond this distance. The size of the simulation box was set more than two times the cut-off distance, corresponding to a supercell size of 1 × 1 × 4 with respect to the size of the conventional cell. To ensure a correct estimate of the uptake, during each GCMC simulation the first 10^7^ steps were used for equilibration followed by 10^7^ production steps for ensemble averages. Each cycle included N attempted Monte Carlo moves; creation, deletion and rotation of the water molecule. The number of N steps at each cycle was equal to the number of water molecules present in the simulation box. To avoid poor sampling at low densities, the number of steps per cycle was set to have a lower limit of 20. All GCMC simulations were performed with the RASPA software package that implements the latest state-of-the-art algorithms for Monte Carlo in various ensembles [50].

## 3. Results and Discussion

### 3.1. Energetics and Optimized Geometries of Monohydrated Functionalized Benzenes from ab initio Calculations

In the present study, a diverse set of 40 functional groups was considered for studying the interaction of their corresponding phenyl derivatives (FBs) with water molecules. The set includes functional groups containing oxygen, such as benzoic acid and phenol, sulfur, nitrogen, phosphorous and halogens. The complete list containing all functionalized phenyl derivatives can be found in Table 1. The examined phenyl derivatives can be further classified as acidic or basic and polar or non-polar according to their functional group. The interpretation of the results follows the incremental binding energies for the water–FB dimers, also presented in Table 1. Table 1 presents the binding energies from RIMP2/def2-TZVPP calculations for the conformers with the most stable configuration for each of the water–FB dimers. As expected, most of the functionalities that were introduced into the phenyl ring increase the binding energy of water with the FBs. The % enhancement of the binding energy reported in Table 1 is a metric of the effectiveness of the functionalization and gives an immediate estimation of the magnitude of the enhancement due to the functionalization with respect to non-functionalized benzene, which—in most cases—is significant and can reach up to 545%. The highest binding energy was calculated for the case of the lithium alkoxide functional group, which was found to be −18.7 Kcal/mol. The magnitude of the binding energy can be attributed to the electrostatic attraction between the positively charged lithium atom of the lithium alkoxide group with the lone electron pair of the oxygen atom of the water molecule. The corresponding optimized geometry is presented in Figure 1. It was observed that the water molecule was located above the plane of the phenyl ring with an O–H bond pointing to the center of the ring. The distance between the lithium atom and the oxygen atom of water was calculated at 1.899 Å, while the distance of the hydrogen atom of the water to the center of the ring was 2.689 Å. The interaction also induced a distortion to the O–H bonds of the water, where the O–H bond towards the ring is elongated (0.978 Å), while the other bond is slightly compressed (0.957 Å). A subset of 15 FBs (columns 2 to 16 in Table 1) follows, with the highest binding energies containing either -OH or -NH_2_ functionalities. The range of the binding energies of a water molecule with these functional groups spans from −12.7 Kcal/mol to −6.4 Kcal/mol, which accounts for more than double the binding energy of a water molecule with benzene. The optimized geometries of the complexes are characterized by the formation of hydrogen bonds between the water molecule and the functional groups. Moreover, depending on the functional group, water can act either as hydrogen donor or hydrogen acceptor. In most cases, except for -O_2_H, -SO_2_CH_3_, -CH_2_NH_2_, -CH_2_OH and -OH, the water molecule forms two hydrogen bonds with the corresponding functional group. The first is formed between the acidic -OH of the functional group and the oxygen atom of water (OH···O_w_), whereas the second is formed between the hydrogen atom of water with the oxygen or nitrogen atom of the functional group (OH_w_···O or OH_w_···N). Among the 15 FBs in this subset, -OSO_3_H has the largest binding energy (−12.7 Kcal/mol) with water forming two hydrogen bonds; the first is formed between the H atom of the -OH group of the functional group with the oxygen atom of water (OH···O_w_) with an H···O_w_ distance of 1.619 Å. The OH bond is slightly elongated by 0.038 Å upon interaction with water, while the second hydrogen bond formed between the hydrogen atom of water and the nearby oxygen atom of the functional group is strained at an OH_w_···O distance of 2.295 Å. Based on the hydrogen bond distances, it is evident that in the case of OH···O_w_ the H-bond is stronger than in the case of OH_w_···O. In addition, the water molecule is located on top of the center of the benzene, with one of the two O–H bonds pointing to the center of the ring, an orientation that further stabilizes dimer formation. Interestingly, both O–H bonds of water are elongated at 0.965 Å. Like in the case of the -OSO_3_H FB, water forms two H-bonds with -PO_3_H_2_, -SO_3_H, -SO_2_H, -COOH, -CNH_2_NOH, -CHNOH, -CONH_2_, -SONH_2_ and -C(OH)_3_ FBs. The corresponding optimized geometries are presented in Figure 1 and selected geometrical parameters of the interaction are summarized in Table 2. It was found that the OH···O_w_ H bond between the acidic proton of the functional group and the oxygen of water is stronger than the OH_w_···O bond (OH_w_···N for -CHNOH), based on the corresponding distances of the -PO_3_H_2_, -SO_3_H, -SO_2_H, -COOH and -CHNOH FBs. Thus, upon interaction with these groups, water acts as hydrogen acceptor rather than hydrogen donor. We also noted that the elongation (Δr) of the O–H bond of the functional group increased as a function of the binding energy, which implies a strong correlation between these two observables. For -CNH_2_NOH FB, water adopts a configuration where two almost equal OH_w_···N (1.981 Å) and OH···O_w_ (1.964 Å) H bonds are formed with -NOH. This reveals that water prefers to form two H-bonds with –ΝOH rather than forming a single H-bond with the lone pair of the nitrogen in the -NH_2_ group, thus acting as hydrogen donor. Similarly, the same behavior is observed for the optimized geometries of the -CONH_2_ and -SONH_2_ groups, which also contain -NH_2_. For these groups, water is more likely to act as a hydrogen donor to the oxygen atoms of the C=O and S=O groups, respectively, while also forming a weak H-bond with its oxygen atom and one of the protons of the amine group.

As mentioned earlier, there are five FBs (-SO_2_CH_3_, -CH_2_NH_2_, -O_2_H, -CH_2_OH and -OH) among the 15 FB subset that form a single H-bond with water, with binding energies ranging from −7.4 Kcal/mol to −6.4 Kcal/mol. These binding energies are the lowest that were calculated for the specific subset of FBs. In water dimers with -SO_2_CH_3_, -CH_2_NH_2_ FBs, water acts as H-bond donor and forms OH_w_···O and OH_w_···N bonds (both at 1.906 Å) with the oxygen and the nitrogen atoms of the -SO_2_ and -NH_2_ groups, respectively. In contrast, water acts an H-bond acceptor in the case of -O_2_H, -CH_2_OH and -OH groups, forming OH···O_w_ H-bonds, the distances of which are calculated to be 1.843 Å, 1.971 Å and 1.856 Å, respectively. It should be noted that in the case of the -CH_2_OH group, water is located above the plane of the ring (almost on top of the carbon of the ring where the functional group is attached) with one OH bond facing towards the center of the ring, similar to the case of -OSO_3_H functionalized benzene.

The second subset of functionalized benzenes consists of 25 FBs (columns 17 to 41 in Table 1), including the reference benzene structure; the optimized geometries of the corresponding dimers are illustrated in Figure 2. Selected geometrical parameters of the optimized geometries are presented in Table 1. In most of the optimized geometries of the dimers, water is located near the functional group of the corresponding FB, but it also adopts a co-planar orientation with respect to the plane of the benzene ring. The exceptions are identified as -NH_2_, -OC_2_H_5_, -OCH_3_, -N≡C and -O_2_CH_2_ groups, groups for which water is located above the plane of the benzene ring. In the cases of benzene and toluene (-CH_3_ group), water can be found above the center of the phenyl ring with one of the OH bonds facing towards the center of the ring (Hw···COM_ring_ distance at 2.44 Å). Regarding the H-bonds formed between water and the compounds of the second subset, a single OH_w_···X (where X stands for the atom that interacts with the proton of the water, for instance the O atom in -OCH_3_ or the N atom in -NH_2_) is formed in most cases. For the -SH FB, water acts as a hydrogen acceptor with the proton of the -SH group oriented towards the oxygen atom of the water, forming an SH···O_w_ H-bond the distance of which is calculated to be 2.202 Å. The corresponding binding energy was found to be −3.4 Kcal/mol, which is only 0.5 Kcal/mol more than the binding energy of benzene. This value is almost half with regard to the corresponding value for the -OH group, which can be attributed to the difference in the electronegativity of the sulfur and oxygen atoms in the groups. For -NH_2_ FB, the binding energy of the water molecule was calculated to be −5.1 Kcal/mol, which is approximately 2 to 4 Kcal/mol lower than the corresponding binding of the -NH_2_-containing functional groups that were presented in the first subset of FBs and that also formed different numbers and types of H-bonds. For the second subset, the range of OH_w_···X distances was found to be 1.932 Å to 2.572 Å, while the OH bond of the water which participates in the formation becomes slightly elongated (maximum elongation at 0.013 Å).

### 3.2. GCMC Simulations for H_2_O Uptake in Functionalized Mg-MOF-74-III

Water uptake for the functionalized Mg-MOF-74-III analogs was calculated by performing GCMC simulations at 298 K for the whole relative humidity range (RH), i.e., up to 100%. The selected MOF structure was functionalized with the -OSO_3_H, -CONH_2_ and -C(OH)_3_ groups, with the corresponding functionalized linkers seen in Figure 3. The selection of these functional groups was based on the results of the QM calculations; these functional groups are among the strongest interacting with water, and they exhibit different binding energies, allowing us to explore any underlying effects of the strength of the interaction on the total water uptake.

To accurately describe the non-bonded interactions between water and the functionalized linkers, σ and ε parameters of the Lennard–Jones potentials for the framework atoms were adjusted by employing a fitting procedure to these parameters to reproduce the QM derived potential. The QM potential was obtained by scanning the potential energy surface at various sites along various orientations around the linker. Electronic energy for each point of the potential energy surface was calculated at the same level of theory as previously mentioned for water–FB dimers. Parameters ε and σ were then fitted to reproduce the calculated binding energies. The QM-derived energies and the corresponding fitted curves can be seen in Figure 4. As can be observed in the potential energy curves plotted in Figure 4, the potential parameters of Dreiding were not able to accurately predict either the equilibrium distance or the equilibrium potential energy depth with respect to our ab initio data, confirming the necessity of fitting the parameters in our ab initio data in order to have an accurate representation of the interactions of water with the functionalized linkers during the GCMC simulations.

The fitted Lennard–Jones parameters for the framework atoms corresponding to the functional groups are presented in Table 3 (columns 2–4). The atom types for the atoms of the functional groups are depicted in Figure 5. In the case of the functionalized MOF-74-III, the Lennard–Jones parameters reported in the first column of Table 3 were used for the framework atoms and were not explicitly treated during the fitting procedure.

The water uptake isotherms for the parent and the three modified frameworks can be seen in Figure 6. At low RH values, the examined materials show low water adsorption until a certain point, after which the water molecules condense and the maximum water uptake is reached. The condensation point varies depending on the structure and the resulting interactions; the stronger the interactions the earlier this point appears, a behavior in agreement with Figure 4. The parent material shows this condensation later than the modified structures, a behavior that can be attributed to the fact that all modified structures show enhanced interactions with water compared with the unmodified structures, thus the condensation appears earlier. Especially for the -C(OH)_3_-modified MOF structure, this point appears in lower RH values than in all other candidates examined. Taking a closer look at the orientation of the functional groups with respect to the other atoms of the linker, it can be seen that sites that experience the binding effect of more than one functional group appear, hence the interactions are stronger and might cause water molecules to condense more easily in the presence of certain functional groups. This can be attributed to the formation of hydrogen bonds of equal strength between the water molecules and the three O–H groups, while for the rest of the functional groups, only one O–H (or N–H) exists, and the additional water molecules occupy the binding sites around the functional group corresponding to lower average binding energies. The optimized geometry from the ab initio calculations corresponding to three water molecules interacting with the functionalized benzene can be seen in Figure 7, where each water molecule forms two hydrogen bonds with the OH functionalities thus acting both as H-donor and H-acceptor. At the maximum loading limit, the volumetric uptake appears more or less the same for the different framework structures studied. This is no surprise, as at this limit, the water capacity of the material depends on the free volume available for the guest molecules to occupy rather than on the strength of the interaction sites. Since all structures come from the same parent material, it is expected that the maximum loading is of approximately the same value. On the contrary, gravimetric uptake depends on the weight of each framework. At the high loading limit, although all structures adsorb almost the same number of water molecules, the gravimetric uptake is higher for the unmodified structures than for the modified candidates. This is due to the fact that functional groups introduce extra weight to the material, thus resulting in these differences.

Figure 8 shows representative snapshots taken at 10% and 60% RH for the parent and the –OSO_3_H functionalized framework. At 10% RH, as also evident from the adsorption curves, the modified material hosts considerably more water molecules than the parent structure. The enhancement of the uptake can be attributed to the stronger binding sites introduced to the structure with the functionalization strategy. That is also the reason why for the –OSO_3_H-modified MOF candidate water molecules are closer to the functional group rather than occupy the empty space. At high water loadings, the uptake tends to increase due to bulk interactions; in snapshots of 60% RH, the free volume of both structures is all almost equally filled. It is worth noting that the existence of 1D channels of hexagonal shape, as well as the type of the linker, are quite important for achieving high water loadings and increasing the hydrophilicity of the functionalized MOF-74-III. The linker in MOF-74-III allows the orientation of the functional groups towards the center of the pore and, consequently, accommodates the water molecules in the most favorable configuration around the functional group in a manner equivalent to the one in the configurations shown in Figure 1 for a single water molecule. However, although the existence of the functional group can partially prohibit the interaction of the water molecules with the benzene rings of the linkers due to steric repulsions, this effect is of limited importance due to the weak interaction of water with the π system of the benzene ring. The functionalization of the linkers can have the same effect on the water uptake in other MOFs with different pore sizes and shapes given that there are no steric effects to prohibit the formation of the water clusters around the functional groups.

## 4. Conclusions

In this work, a comprehensive computational study on the interaction of water with functional groups was performed by employing ab initio calculations and Monte Carlo simulations. A large set of 40 functional groups was considered, comprised of groups that can be hydrophilic or hydrophobic in nature. The interaction energies and the corresponding optimized geometries for the water—functionalized benzene dimers were calculated at the RIMP2/def2-TZVPP-level of theory for all the functional groups considered. The obtained results revealed that the interaction energies of the water molecule with the functionalized benzenes can be increased significantly and can be as high as −18.7 Kcal/mol in the case of the -OLi group. Other functional groups such as -OSO_3_H and -PO_3_H_2_ also showed high interaction energies with water, which were calculated to be at −12.7 and −11.8 Kcal/mol, respectively, more than four times larger than the interaction energies of water with benzene. Among the functional groups considered, the largest interaction energies were calculated for those water—functionalized benzene dimers where water can simultaneously form two hydrogen bonds with the corresponding functional groups, a direct hydrogen bond between the hydrogen atom of the functional group and the oxygen atom of the water, and a strained hydrogen bond between the oxygen (or nitrogen) atom of the group and one of the hydrogen atoms of the water. The magnitude of the interaction energy is directly related to the hydrophilicity of the functional group, which is known to have significant impact on the water adsorption behavior of functionalized porous materials. To test the effect of the incorporation of the functional groups on the water adsorption properties of the MOFs, GCMC simulations were conducted on the pristine and the functionalized Mg-MOF-74-III at room temperature and relative humidity up to 100%. The selection of Mg-MOF-74-III as a platform for our study was based on the uniform 1D pore system, the pore size near the critical diameter for water (~20 Å) and the ability of the organic linker to be functionalized. The calculated isotherms for the functionalized Mg-MOF-74-III with hydrophilic functional groups, such as -OSO_3_H, -CONH_2_ and -C(OH)_3_, showed that their introduction onto the organic linker had a strong impact on adsorption behavior, causing a sharp increase in the water uptake values and reaching saturation at lower % RH with respect to the parent MOF. This is directly related to the enhanced hydrophilicity of the MOF due to the existence of the specific functional groups in the pores with respect to the hydrophobic -CH_3_ group of the pristine material. The hydrophilic functional groups act as additional adsorption sites for water, complementary to the open metal sites that are present in this MOF. The saturation capacity of the pristine and functionalized Mg-MOF-74-III are among the highest reported to date for MOF materials [21,25]. The volumetric and gravimetric saturation capacity was found to be 850 cm^3^/cm^3^ and 72.5 mmol/g, respectively, for the pristine Mg-MOF-74-III. For the functionalized derivatives, the gravimetric saturation uptake was found to be reduced depending on the additional mass added in the MOF structure after the introduction of the heavier functional groups with respect to the pristine MOF; however, the uptake remained among the highest reported, signifying a compromise between maximum gravimetric capacity and water affinity at low % RH. The effect of the functionalization of organic linkers can undoubtedly influence the water adsorption behavior of MOFs, but we cannot anticipate that the observed effects in this study can be universally true for most MOFs due to the large diversity of the topology, pore size distribution, pore shape and dimensionality. Although the organic linker functionalization technique is proven to be an effective strategy, more studies are needed for a variety of different MOF structures in order to be able to design tailor-made functionalized materials with the desired water adsorption properties for specific applications, which we will address in future publications.

## Figures and Tables

**Figure 1 molecules-27-02614-f001:**
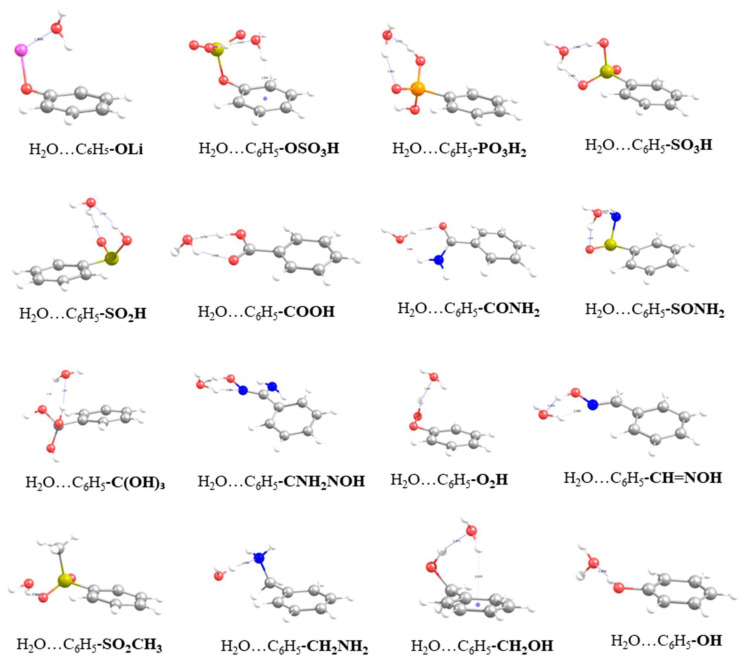
Optimized geometries at the RIMP2/def2-TZVPP-level of theory for the first subset of 16 functionalized benzenes.

**Figure 2 molecules-27-02614-f002:**
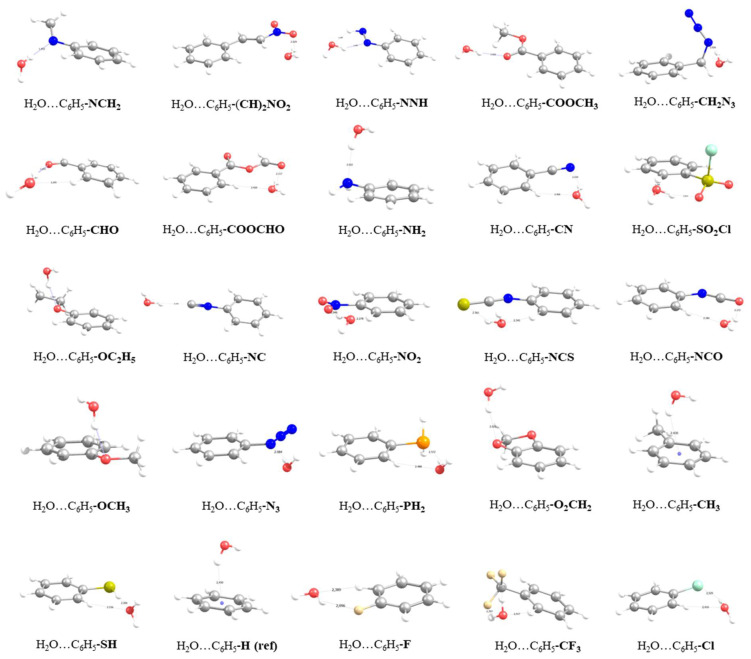
Optimized geometries at the RIMP2/def2-TZVPP-level of theory for the second subset of 25 functionalized benzenes.

**Figure 3 molecules-27-02614-f003:**
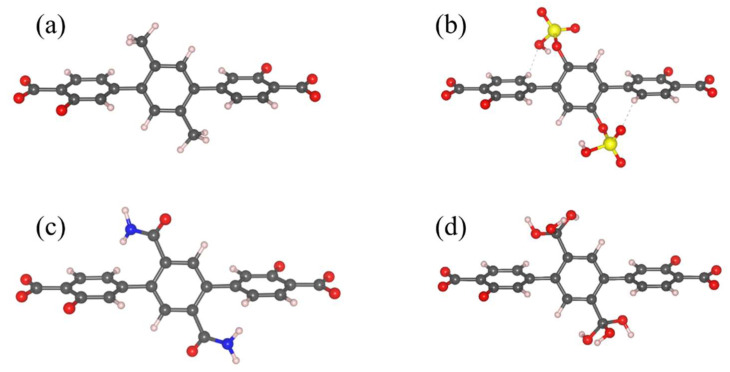
The functionalized linker structure of Mg-MOF-74-III considered in the GCMC simulations; the original Mg-MOF-74-III linker (**a**), the –OSO_3_H (**b**), –CONH_2_ (**c**) and –C(OH)_3_ (**d**) functionalized linker. Carbon, hydrogen, oxygen, sulfur and nitrogen atoms are depicted as grey, pink, red, yellow and blue spheres, respectively.

**Figure 4 molecules-27-02614-f004:**
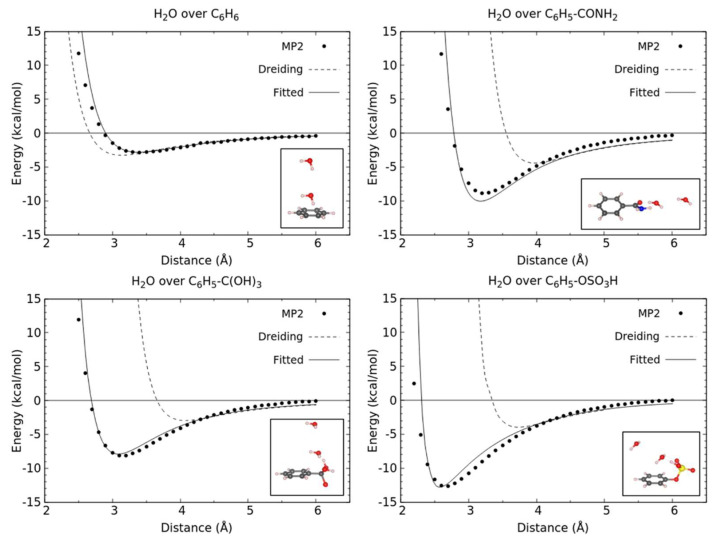
Comparison between the interaction energy curves for H_2_O molecules around the parent and modified linkers of this study. Interaction energies were calculated using MP2 (filled symbols), unfitted Lennard–Jones potentials (dashed line) and fitted Lennard–Jones potentials (solid lines). The inset figure in each case shows the orientation along which the binding energies were calculated.

**Figure 5 molecules-27-02614-f005:**
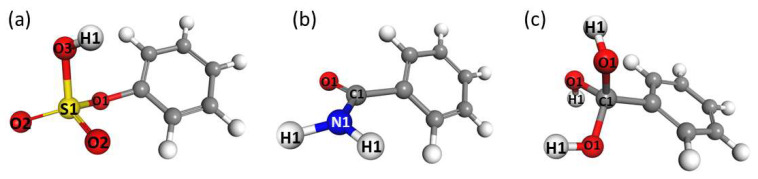
The different atom types of the atoms of the functional groups that were considered in the force field fitting procedure, (**a**) -OSO_3_H, (**b**), -CONH_2_ and (**c**) -C(OH)_3_.

**Figure 6 molecules-27-02614-f006:**
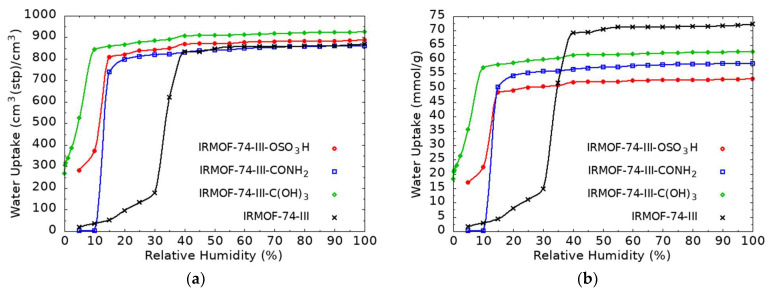
Volumetric (**a**) and gravimetric (**b**) water uptake for the parent and the functionalized Mg-MOF-74-III.

**Figure 7 molecules-27-02614-f007:**
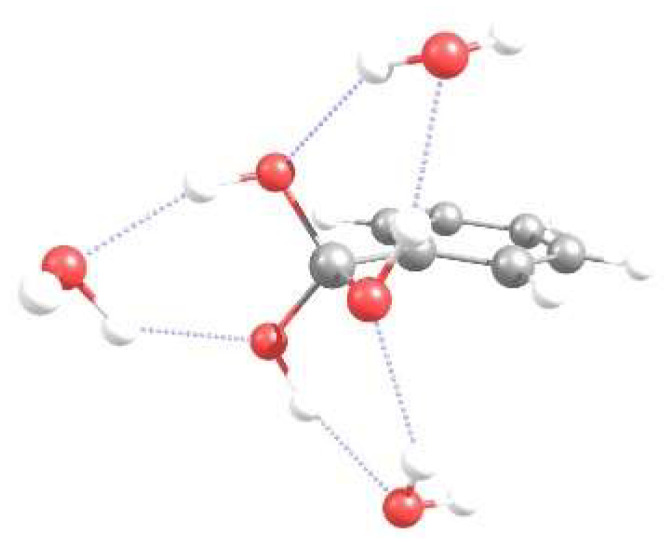
Optimized geometry corresponding to three water molecules interacting with the -C(OH)_3_ functionalized benzene, calculated at RIMP2/def2-TZVPP-level of theory. Dashed lines indicate the hydrogen bonds formed between water and the functional group.

**Figure 8 molecules-27-02614-f008:**
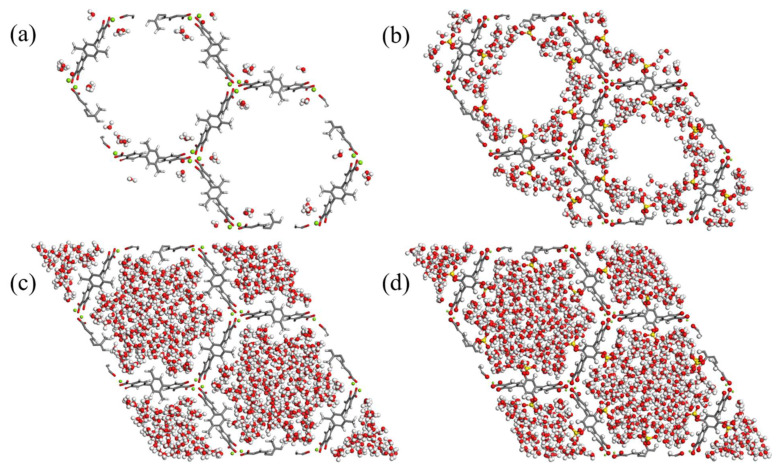
Snapshots taken from the GCMC simulations for the parent MOF-74-III material at 10% RH (**a**) and 60% RH (**c**) and for the –OSO_3_H functionalized at 10% RH (**b**) and 60% RH (**d**).

**Table 1 molecules-27-02614-t001:** Counterpoise-corrected interaction energies in Kcal/mol between the H_2_O molecule and the functionalized benzenes, obtained at the RI-MP2/def2-TZVPP level.

	Functional Group	B.E. (Kcal/mol)	% Enhancement
1	-OLi	−18.7	545%
2	-OSO_3_H	−12.7	338%
3	-PO_3_H_2_	−11.8	307%
4	-SO_3_H	−11.0	279%
5	-SO_2_H	−10.1	248%
6	-COOH	−9.4	224%
7	-CONH_2_	−8.9	207%
8	-SONH_2_	−8.3	186%
9	-C(OH)_3_	−8.2	182%
10	-CNH_2_NOH	−8.1	179%
11	-O_2_H	−7.4	155%
12	-CHNOH	−7.1	145%
13	-SO_2_CH_3_	−6.9	138%
14	-CH_2_NH_2_	−6.9	138%
15	-CH_2_OH	−6.5	124%
16	-OH	−6.4	121%
17	-NCH_2_	−5.9	103%
18	-(CH)_2_NO_2_	−5.7	97%
19	-N=NH	−5.6	93%
20	-COOCH_3_	−5.5	90%
21	-CH_2_N_3_	−5.4	86%
22	-CHO	−5.4	86%
23	-COOCHO	−5.4	86%
24	-NH_2_	−5.1	76%
25	-C≡N	−4.9	69%
26	-SO_2_Cl	−4.9	69%
27	-OC_2_H_5_	−4.4	52%
28	-N≡C	−4.4	52%
29	-NO_2_	−4.1	41%
30	-N=C=S	−4.1	41%
31	-N=C=O	−4.0	38%
32	-OCH_3_	−3.9	53%
33	-N_3_	−3.7	28%
34	-PH_2_	−3.5	21%
35	-O_2_CH_2_	−3.5	21%
36	-CH_3_	−3.4	17%
37	-SH	−3.4	17%
38	-H	−2.9	0% (ref)
39	-F	−2.9	0%
40	-CF_3_	−2.7	−7%
41	-Cl	−2.6	−10%

**Table 2 molecules-27-02614-t002:** Summary of the geometrical parameters of the interaction of the water molecule with the functionalized benzenes obtained at the RI-MP2/def2-TZVPP-level. Binding energies (B.E.), bond lengths and bond angles are in Kcal/mol, Å and degrees (°) respectively.

	Functional Group	B.E., Kcal/mol	OH···Ow, Å	OHw···O(N), Å	Ow-H-O(N), °	Ow-Hw-O(N), °	Δr OH, Å	H-Ow (1), Å	H-Ow (2), Å	Hw-COM ring, Å
1	-OLi	−18.7						0.957	0.978	2.689
2	-OSO_3_H	−12.7	1.619	2.295	169.453	122.289	0.03849	0.965	0.964	2.509
3	-PO_3_H_2_	−11.8	1.733	1.862	156.068	146.685	0.02433	0.976	0.958	
4	-SO_3_H	−11.0	1.688	1.983	162.974	133.704	0.02661	0.969	0.959	
5	-SO_2_H	−10.1	1.759	1.956	153.22	142.999	0.02132	0.973	0.96	
6	-COOH	−9.4	1.751	1.928	158.802	140.045	0.01921	0.972	0.959	
7	-CONH_2_	−8.9	2.006	1.849	140.504	153.63	0.00605	0.974	0.958	
8	-SONH_2_	−8.3	2.075	1.87	140.482	151.84	0.00421	0.973	0.958	
9	-C(OH)_3_	−8.2	1.899	2.104	151.809	104.017	0.012	0.966	0.96	
10	-CNH_2_NOH	−8.1	1.964	1.981	140.17	140.17	0.00933	0.971	0.958	
11	-O_2_H	−7.4	1.843		157.52		0.01273	0.963	0.96	
12	-CH=NOH	−7.1	1.924	2.086	143.1	128.61	0.01028	0.967	0.958	
13	-SO_2_CH_3_	−6.9		1.906		153.78		0.968	0.958	
14	-CH_2_NH_2_	−6.9		1.906		164.72		0.976	0.958	
15	-CH_2_OH	−6.5	1.971		158.707		0.00639	0.963	0.959	2.619
16	-OH	−6.4	1.856		175.49		0.00917	0.959	0.959	
17	-NCH_2_	−5.9		1.952		167.84		0.971	0.958	
18	-(CH)_2_NO_2_	−5.7		2.029		151.56		0.965	0.958	
19	-N=NH	−5.6	2.299	1.995	127.693	141.95	0.0005	0.969	0.958	
20	-COOCH_3_	−5.5		1.932		169.59		0.966	0.957	
21	-CH_2_N_3_	−5.4		2.036	147.784			0.966	0.958	
22	-CHO	−5.4		1.933		166.16		0.966	0.957	
23	-COOCHO	−5.4		2.117		167.73		0.964	0.958	
24	-NH_2_	−5.1		2.022		165.24		0.967	0.958	
25	-C≡N	−4.9		2.23		148.7		0.963	0.958	
26	-SO_2_Cl	−4.9		2.014				0.963	0.958	
27	-OC_2_H_5_	−4.4		1.941		162.81		0.965	0.958	
28	-N≡C	−4.4		2.165		178.45		0.965	0.957	
29	-NO_2_	−4.1		2.046		156.07		0.962	0.958	
30	-N=C=S	−4.1		2.561		159.05		0.963	0.959	
31	-N=C=O	−4.0				138.39		0.962	0.958	
32	-OCH_3_	−3.9		1.983	162.152			0.964	0.958	
33	-N_3_	−3.7		2.084		158.62		0.964	0.958	
34	-PH_2_	−3.5		2.572		151.67		0.964	0.958	
35	-O_2_CH_2_	−3.5		2.021		153.58		0.963	0.958	
36	-CH_3_	−3.4						0.962	0.959	2.43
37	-SH	−3.4	2.206			172.98	0.0014	0.959	0.959	
38	-H	−2.9						0.961	0.958	2.45
39	-F	−2.9		2.096		148.47		0.96	0.958	
40	-CF_3_	−2.7						0.96	0.959	
41	-Cl	−2.6	2.416	2.529				0.961	0.958	

**Table 3 molecules-27-02614-t003:** Lennard–Jones parameters for the framework atoms corresponding to parent (-CH_3_) and functionalized Mg-MOF-74-III.

-CH_3_ (Parent Mg-MOF-74-III)			-OSO_3_H			-CONH_2_			-C(OH)_3_		
	ε (Κ)	σ (Å)		ε (Κ)	σ (Å)		ε (Κ)	σ (Å)		ε (Κ)	σ (Å)
C-Ow	61.09	3.51	O1-Ow	61.29	3.09	C1-Ow	61.09	3.51	C1-Ow	47.85	3.87
O-Ow	61.29	3.09	O2-Ow	61.29	2.69	O1-Ow	61.29	3.09	O1-Ow	48.16	2.22
H-Ow	66.01	3.13	O3-Ow	61.29	2.69	N1-Ow	55.12	3.21	H1-Ow	7.65	0.65
Mg-Ow	24.43	3.20	S1-Ow	116.2	3.27	H1-Ow	24.43	2.40			
			H1-Ow	24.43	1.92						

## Data Availability

The data that support the findings of this study are available from the corresponding author upon reasonable request.

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
