# Peer review of "Linker Functionalization Strategy for Water Adsorption in Metal–Organic Frameworks"

_molecules, 2022, doi:10.3390/molecules27092614_

Round 1
Reviewer 1 Report
In this manuscript Giappa et al. present a computational study examining the interaction of water with functional groups on benzene containing organic linkers used in MOF synthesis. Screening calculations of binding energy for several functional groups are presented in tandem with GCMC simulations of water adsorption isotherms on select functional groups within Mg-MOF-74-III. Overall the data presented in this manuscript has the potential to be impactful if the discussion and impact of the results can be improved, correlated with one another and justified. Considering this assessment, this manuscript may be accepted for publication in Molecules after major revisions. Some comments to assist with this revision that can be addressed are noted below as follows.
- How do the optimized geometries of one functionalized linker interacting one water molecule presented in Figure 1 and Figure 2 correlate with bulk adsorption interaction of MOF structure with many more water molecules as in GCMC simulation? Is this translatable for other MOF structures? If questionable, this should be emphasized.
- Is the bulk uptake impacted by bond orientation considering the Mg-MOF-74-III structure has hexagonal channel pores? The discussion regarding bond distances and orientation of H2O within and outside of benzene ring plane seems disjointed from other results presented in the manuscript.
- Line 173: QM is not defined.
- Line 303: The “% enhancement” column in Table 1 is not mentioned in the discussion. Authors should describe the impact of this calculated value.
- Line 305: Table 2 is missing units in the column headers for the values shown. It is hard to interpret which values are lengths vs. angles vs. energies.
- Line 314: What is the reasoning for selecting these three specific functionalities for the GCMC simulations? The results of BE alone from Table 1 would imply that the top 3 strongest binding energies are -OLi, -OSO3H, and -PO3H2 instead of the three selected for GCMC simulations. Additional justification for this selection should be presented.
- Line 334: The discussion of observed results from Figure 4 is missing. What is the conclusion from this interaction energy comparison? What do the differences between “MP2” and “Dreiding” data mean?
- Line 338-339: Authors show GCMC simulation of water uptake in isotherms shown in Figure 5. The validity and reliability of this simulation (and the corresponding model parameters) should be compared to corresponding experimental data from literature for at least the parent Mg-MOF-74 at 298K. This would strengthen the simulation data and justify the differences observed.
- Line 346-347: The “condensation point” appears for the -C(OH)3 modified structure at lower RH than the other structures. This contradicts the differences observed in binding energies reported in Table 1. Authors should comment on this discrepancy and provide justification for why this is the case. Authors mention about orientation of functional group that may require more elaboration to tie in with binding energy and condensation point.
Reviewer 2 Report
In the manuscript, a large set of functionalized benzenes (FBs) for water adsorption were simulated and then examined in MOF-74. Both ab-initio calculations and Monte Carlo simulations were applied in the study of the interaction of FBs with water and then the water uptake as a function of relative humidity (RH). The hydrophilicity were greatly increased for the chosen substituents as well as volumetric water uptakes, among the best reported MOF candidates, shedding lights on the linker-modified MOFs. Thus I suggest the manuscript to be accepted with minor changes.
- In the manuscript, the terminology should be written as full name when it first come up, like the RH (relative humidity) in line 21.
- In line 117, several criteria were mentioned of selecting the MOF candidate. But the strength of MOF-74-III fitting these criteria were not addressed further.
- In part 3.2, GCMC simulations for H2O uptake in functionalized Mg-MOF-74-III, the water uptakes for different substituted MOF-74 were calculated. Are there any experimental data cohering with the simulations? Are those FBs synthesized by the researchers or ever reported?
- To calculated the volumetric water uptake of different MOF-74, how does the researchers consider the open metal sites and the pore size shrink when they have large substituents like sulfonic acid and phosphonic acid.
Reviewer 3 Report
Recommendation: Publish after minor revision.
Comments:
In this work, the authors studied the water adsorption behavior in functionalized Mg-MOF-74-III by employing ab initio calculations and Monte Carlo simulations. It is potentially would be helpful for designing the MOF materials with high water capture. The following issues should be addressed before accepted for publication.
- The fitted parameters for the Lennard-Jones potential should be listed in the Section of “Results and discussion”.
- 3.2, why the authors chose -OSO3H, -CONH2 and -C(OH)3? It seems the interaction between water and some functionalized groups listed in Table 1 (i.e., 1-8) are stronger than -C(OH)3.
- It is a bit confused to me for Figures 5 and 6. It looks the number of water molecules in Fig. 6d is significantly larger than that of Fig. 6b. But from Fig.5a, the number of water uptake with RH=60% in both systems are close. I recommend the authors add a figure to shown the numbers of water molecules as the function of RH.
Reviewer 4 Report
The paper entitled “Linker functionalization strategy for water adsorption in Metal Organic Frameworks” reported an interesting work on the very important topic of MOFs potential to be used in adsorption-based water capture. The article is well written, summarizes a lot of valuable information in its figures and tables, and contains a lot of work. The manuscript needs several revisions before it can be published. Therefore, please improved/clarified the following points:
- Please describe the Mg-MOF-74-III. Where is III coming from?
- Lattice parameters, bond lengths, and angles for Mg-MOF-74-III should be calculated.
- Are there some examples in the literature to compare the computational results with? It is necessary to insert a table in this respect.
- The homogeneity of the reference section needs to be maintained. Some references are abbreviated, while others are not or are abbreviated incorrectly (e.g., refs 20, 23, 24). Please check and revise the whole reference section according to the Molecules instructions for authors (https://www.mdpi.com/journal/molecules/instructions)
Based on these, I advise the authors to rectify the above mentioned errors, and I hope to re-evaluate the revised manuscript.
Round 2
Reviewer 1 Report
The authors have adequately addressed the comments provided in the original review. The revised version of the manuscript may be published to Molecules.
Reviewer 4 Report
Congratulations on a great job. The author has made substantial improvements to this article. The manuscript can be accepted for publication in the present form.